# Intra-Sample Heterogeneity of Potato Starch Reveals Fluctuation of Starch-Binding Proteins According to Granule Morphology

**DOI:** 10.3390/plants8090324

**Published:** 2019-09-04

**Authors:** Stanislas Helle, Fabrice Bray, Jean-Luc Putaux, Jérémy Verbeke, Stéphanie Flament, Christian Rolando, Christophe D’Hulst, Nicolas Szydlowski

**Affiliations:** 1Univ. Lille, CNRS, USR 3290—MSAP—Miniaturisation pour la Synthèse, l’Analyse et la Protéomique, F-59000 Lille, France; 2Univ. Lille, CNRS, UMR 8576—UGSF—Unité de Glycobiologie Structurale et Fonctionnelle, F-59000 Lille, France; 3Florimond Desprez Veuve-et-Fils Ind., Section Biotechnologies, BP 41, 59242 Cappelle en Pévèle, France; 4Univ. Grenoble Alpes, CNRS, CERMAV, F-38000 Grenoble, France

**Keywords:** starch granule, starch, potato, *Solanum tuberosum*, proteomics, amylopectin, amylose

## Abstract

Starch granule morphology is highly variable depending on the botanical origin. Moreover, all investigated plant species display intra-tissular variability of granule size. In potato tubers, the size distribution of starch granules follows a unimodal pattern with diameters ranging from 5 to 100 µm. Several evidences indicate that granule morphology in plants is related to the complex starch metabolic pathway. However, the intra-sample variability of starch-binding metabolic proteins remains unknown. Here, we report on the molecular characterization of size-fractionated potato starch granules with average diameters of 14.2 ± 3.7 µm, 24.5 ± 6.5 µm, 47.7 ± 12.8 µm, and 61.8 ± 17.4 µm. In addition to changes in the phosphate contents as well as small differences in the amylopectin structure, we found that the starch-binding protein stoichiometry varies significantly according to granule size. Label-free quantitative proteomics of each granule fraction revealed that individual proteins can be grouped according to four distinct abundance patterns. This study corroborates that the starch proteome may influence starch granule growth and architecture and opens up new perspectives in understanding the dynamics of starch biosynthesis.

## 1. Introduction

Starch is the main source of calories in human and livestock diets. It is also used in glue, bioplastic, and biofuel industries as well as a food additive. Starch is composed of two glucose polymers, amylose and amylopectin, and accumulates in the form of cold water-insoluble granules. Amylose and amylopectin are both constituted by α-1,4 glucan linear segments ramified through α-1,6 *O*-glycosidic bonds (also called branching points) [1]. However, while amylose is mainly linear with less than 1% of α-1,6 linkages, amylopectin is moderately branched and contains 5 to 6% of α-1,6 bonds [2,3]. Starch also contains minor compounds such as proteins [4], most of which being starch metabolizing enzymes [5].

Starch synthesis involves several families of enzymes. Starch synthases (SSs) are responsible for the elongation of α-1,4 glucans [6]. On the one hand, the SS1 isoform synthesizes the shortest amylopectin chains [7]. On the other hand, SS2 synthesizes medium-size chains and SS3 elongates the longest chains that compose amylopectin molecules [8,9]. A fourth isoform, SS4, is involved in the initiation of starch synthesis as illustrated by the mutant phenotypes of Arabidopsis plants [10,11,12,13,14]. Two other isoforms of SSs, SS5 and SS6 were found in the potato genome [15]. Moreover, in a previous study, we identified these proteins in association with potato starch by proteomics based on mass spectrometry [16]. However, their putative functions in starch synthesis remain unknown. The granule-bound starch synthase (GBSS) elongates amylose glucans and is noticeably the most abundant protein bound to the starch granule. The starch branching enzymes (SBE1 and SBE2) catalyze the formation of α-1,6 linkages in starch polymers after rearranging a pre-existing α-1,4 glucan [17]. Both of them were found in association with the potato starch granule in our previous study [16]. Moreover, we showed the presence of a third isoform annotated as BE3. Further sequence analysis revealed that this enzyme is a variant of BE1. Both enzymes were thus renamed BE1.1 and BE1.2, respectively. Debranching enzymes (DBEs) hydrolyze α-1,6 linkages. Among them, isoamylase 1 and 2 (ISA1 and ISA2) are involved in amylopectin synthesis, whereas isoamylase 3 (ISA3) participates in starch degradation [18,19]. Among starch-synthesizing enzymes, GBSS, SS1, SS2, SS3, BE1, BE2, ISA1, and ISA2 were found in association with amyloplastic starch in several species such as potato, maize, rice, pea, barley, and wheat [20,21,22,23,24,25,26,27] and ISA 3 was recently identified in potato starch [16].

Other proteins observed in association with the starch granule are involved in starch degradation and its regulation. The glucan water dikinase (GWD) and the phosphoglucan water dikinase (PWD) add phosphate groups at C6 or C3 position of the glucose residues, respectively, to initiate starch degradation [28,29]. The phosphatases starch excess 4 (SEX4) and like SEX four 2 (LSF2) were also identified as chloroplastic proteins and possess a carbohydrate binding module (CBM) [30,31]. SEX4 is able to dephosphorylate phosphoglucans in C3 and C6 positions, whereas LSF2 is only active on C3-phosphoesters [31]. Moreover, SEX4 was observed in the surrounding of starch granules by fluorescent microscopy and LSF2 peptides were identified by SDS-PAGE LC-MS/MS analysis of Arabidopsis starch-bound proteins [31,32]. Recently, other proteins were found associated to the starch granule in Arabidopsis leaves [33,34,35]. These include protein targeting to starch 1 (PTST1), which addresses GBSS to the granule, early starvation protein 1 (ESV1), and its homologue LESV [33,34]. The two latter proteins are involved in the control of starch degradation. It was proposed that both proteins modulate the molecular organization of starch and consequently affect the glucan accessibility to catabolic enzymes. Noteworthy, we recently confirmed the presence of these proteins in potato starch with the addition of the starch phosphorylases PHS1a, PHS1b, and PHS2 [16]. Furthermore, our study revealed the presence of a thioredoxin (TRx), a gluthatione perodydase (GPx) and a peptidyl-prolyl-cis-trans isomerase (CYP20-20). On the one hand, the Arabidopsis counterpart of THRx is involved in the ferredoxin/thioredoxin pathway that regulates metabolic processes in function of light and is known to activate SS1 and the beta-amylase BAM1 [36,37]. On the other hand, glutathione peroxidases are thought to contribute to cellular redox homeostasis during plant-cell stress responses [38]. Moreover, the relative abundance of starch-associated proteins is likely under the control of regulatory mechanisms related to the initiation and biogenesis of the granules [39].

Starch granules display a broad structural and morphological eterogeneity. This disparity is observed at the level of: (i) the botanical origin; (ii) the organelle and tissue localization within a given species; (iii) the individual cells of a given tissue; (iv) within the same plastid, where distinct granule populations can be observed [40,41]. In potato starch, a high diversity of granule morphology can be observed with diameters ranging from 5 to 100 µm [42]. A previous study indicated intra-sample variations of potato starch structure and composition with large granules displaying higher amylose contents and an increased proportion of long amylopectin chains (≥ DP 37) [40]. Moreover, in the latter study, the smallest granules displayed a higher proportion of small amylopectin chains (DP 6-13) and an A-type X-ray diffraction pattern [43]. In order to evaluate the potential heterogeneity of starch-binding proteins in link with starch structure and morphology, we investigated the size and shape repartition as well as the composition and structure of size-fractionated potato starch granule populations with average diameters of 14.2 ± 3.7 µm, 24.5 ± 6.5 µm, 47.7 ± 12.8 µm, and 61.8 ± 17.4 µm (fractions A, B, C, and D, respectively). Amylose content, starch crystallinity, amylopectin chain length distribution, and phosphate contents were determined as well as the protein content and stoichiometry. In addition to changes in phosphate content as well as small differences of the amylopectin structure, we found that the starch-binding protein stoichiometry significantly varies according to granule diameter. Quantitative proteomics revealed that individual proteins can be grouped according to four distinct abundance patterns indicating that the association between glucans and individual proteins might be specifically regulated.

## 2. Results

### 2.1. Heterogeneity of Starch Morphology and Size-Fractionation

Starch granules isolated from potato (cv. Monalisa) tubers were analyzed for size and shape distribution by granulomorphometry (Figure 1). The granule shape was estimated by measuring the ellipsoid roundness corresponding to the ratio between the average radius of peripheral circles and the radius of the maximal circle included in the particle. The granule size was characterized by the area diameter that corresponds to the diameter of a perfect circle of the same area than the particle. A high heterogeneity of granule size was observed, with diameters ranging from 4 to 100 µm and most of the granules displayed a diameter between 15 and 40 µm (Figure 1a). Starch granules were mostly roundish with an ellipsoid roundness above 60% (Figure 1b). Moreover, the smaller the granule, the higher the roundness. Indeed, most starch granules below 40 µm diameter displayed an ellipsoid roundness above 80%, while bigger particles showed important shape disparity (Figure 1c).

In order to investigate the disparity of starch structure and composition according to the granule morphology, starch was fractionated sequentially with the use of nylon filters (11, 31, and 50 µm mesh widths). Each fraction was then analyzed by granulomorphometry and scanning electron microscopy (Figure 2 and Table 1). Average granule diameters of 14.2 ± 3.7 µm (fraction A), 24.5 ± 6.5 µm (fraction B), 47.7 ± 12.8 µm (fraction C), and 61.8 ± 17.4 µm (fraction D) were measured in the different fractions (Figure 2e and Table 1). As expected from the analysis of unfractionated samples, smaller granules were more spherical with an average ellipsoid roundness of about 80% for fraction A and B, while fraction C and D displayed an ellipsoid roundness about 70% (Figure 2a–d, and Table 1).

### 2.2. Structure and Composition of Starch

A previous study reported on the structural heterogeneity of size-fractionated potato starch granules [43]. This work highlighted differences in the starch X-ray diffraction (XRD) patterns, amylose contents, and the amylopectin chain length distribution (CLD) among three size fractions [43]. To further investigate these disparities, we determined protein, amylose, and phosphate contents as well as the amylopectin structure in fractions A, B, C and D. On the one hand, no significant differences were observed regarding the amylose content, starch crystallinity and the wavelength at the maximum of absorption (λ_max_) of the amylopectin-I_2_ complex (Table 1 and Appendix A). On the other hand, protein contents significantly varied in the different granule fractions with a maximum of 2.8 ± 0.12 µg.mg^−1^ in fraction B (Table 1). However, these little changes did not follow any specific correlation with the average granule diameter of the corresponding fractions. Conversely, phosphate contents were significantly reduced with the increase of the average granule diameter (Figure 3a). A two-fold decrease of the total phosphate (P) content was observed in fraction D compared to fraction A with 1.9 ± 0.3 and 4.2 ± 0.3‰, respectively (Figure 3a). Although both C3- and C6-P contents followed the same pattern, the decrease in total P was mainly the consequence of the reduction of C6-P contents (Figure 3b,c). Hence, the proportion of P at the C3 position increased by 1.13- and -1.38 fold in fractions B and C, respectively, compared to fraction A (Figure 3d). In order to investigate the structural heterogeneity of amylopectin from the different fractions, total starch was debranched and chain length distribution (CLD) was determined by fluorescence-assisted capillary electrophoresis (FACE) (Figure 4). A subtle decrease of glucans from DP 10 to 24 was observed with the increase of granule diameter among the fractions, in the order fractions A, B, C, and D (Figure 4a). This was correlated with an increase in the relative abundance of longer chains, from DP 45 to 100 (Figure 4b).

### 2.3. Quantitative Proteomics and Correlation Analysis

Each fraction was analyzed by shotgun mass spectrometry using 300 mg of starch. Starch granules were washed with 2% SDS to remove unspecifically bound proteins prior to gelatinization and isolation of internally trapped proteins. Peptides were obtained by the eFASP (enhanced filter-aided sample preparation) method and analyzed by UPLC-nano-ESI-MS/MS. Protein quantification was performed with the Maxquant and Perseus 1.6.0.2 software and a custom database to avoid quantification biases because of the redundant gene fragments present in the PGSC (potato genome sequencing consortium) databank (Appendix A). This database included proteins that we previously identified in association with potato starch and comprised most of the starch metabolizing enzymes as well as a thioredoxin (THRx), a gluthatione peroxidase (GPx), and a peptidyl-propyl-*cis*-*trans*-isomerase (CYP20-20) [16].

In order to evaluate protein stoichiometry within starches of the different fractions, the abundance score (iBAQ, intensity based absolute quantification) of each protein was expressed in fmol per mg of starch (Appendix A, Table 1, Figure 5). Regardless of the fraction, GBSS and SS2 remained the two major proteins, directly followed by LSF2 and LESV that were about 10 times less abundant than SS2 (Figure 5). Interestingly, the relative amount of GWD continuously decreased with the increase of granule average diameter, which was consistent with the observed decrease in glucose-6-P content. The relative abundance of the other proteins also displayed significant variations between the four fractions (Figure 5). This was still the case concerning the less abundant proteins including ISA3, SS4, SS6, PHS1a, and PHS1b. To better illustrate the changes in protein abundance, the concentration of individual proteins in fraction A was set to 1 and fold changes were plotted versus the average granule diameter of the different fractions (Figure 6). SS3 and PTST1 displayed fold changes above 4 while other protein concentrations such as those of GBSS or SS4 varied faintly according to granule diameter with less than 0.2-fold variations (Figure 6). Furthermore, SS1, 2 and 3, BE2, LESV, PWD, PTST1, PHS1a and b, and THRx harbored fold changes above 1 between fraction A and the other fractions. Overall, starch-bound proteins could be grouped according to four distinct abundance patterns. SS4, SS6, BE1.1, BE1.2, ISA3, GPx, and CYP20-2 showed similar patterns with a decrease in protein amounts in fraction B followed by a continuous increase in fractions C and D (Figure 6). SS1, 2 and 3, BE2, PTST1, LSF2, ESV1, LESV, PWD, PHS1a and b grouped together, displaying an inverse pattern with an increase in fraction B and C followed by a decrease in fraction D (Figure 6). SEX4 and SEX4-like harbored a unique pattern while GBSS, THRx, and GWD showed faint variations among the fractions (Figure 6). Paired samples Pearson correlation analysis confirmed these protein behaviors with the strongest positive correlation observed between PTST1 and LSF2 (Appendix A). Noteworthy, SS4 and SS6 were strongly anticorrelated with PWD and LESV, illustrating the opposition between their corresponding patterns (Figure 6 and Appendix A).

A correlation analysis was also performed between individual protein concentrations and the other parameters measured in this study (Appendix A). SS1, 2 and 3 showed negative correlations with the relative abundance of amylopectin chains of DP 13 to 28 while their abundance was positively correlated with chains of DP 4 to 10 and 31 to 40 (Appendix A). As expected from the positive correlations between the concentrations of LSF2, BE2, PTST1, SEX4, or SEX4-like and the concentrations of SS1, 2 and 3, the former proteins displayed similar correlations with the abundance of these amylopectin chains (Appendix A). On the other hand, BE1.1, BE1.2, CYP20.2, GPX, and ISA3 showed similar positive correlations with the proportion of the longest amylopectin glucans (above DP 42) (Appendix A). Most strikingly, the abundance of GWD did not correlate with that of any other starch-bound protein (Appendix A). However, it was significantly correlated with the amounts of starch phophoesters, especially at C6 position, as well as with granule morphology and the amylopectin chain length distribution (Appendix A).

## 3. Discussion

We investigated the heterogeneity of structure, composition, and morphology existing within individual starch samples isolated from whole potato tubers. While the morphological heterogeneity of starch is well-known and often relates to the botanical, tissular, and cellular origins [40], variations in starch composition within a starch extract were scarcely investigated. A recent study highlighted that small potato starch granules harbored a facetted morphology with an A-type crystalline structure contrary to the bigger ovoid B-type starch granules [43]. In our study, such differences of crystalline structure could not be observed by XRD analysis. Moreover, while small granules were significantly rounder than bigger ones, a vast majority displayed a spherical or ovoid smooth morphology as observed by SEM and confirmed by granulomorphometric analysis. These discrepancies might originate from the use of different potato cultivars in both studies or from the use of commercial starch that could contain traces of B-type starch from other species. However, the origin of the starch used in the work from Wang and collaborators was not indicated [43]. On the other hand, we observed similar variations of the amylopectin chain length distribution according to the average granule diameter. The relative proportion of glucans from DP 10 to 24 decreased with the increase of granule diameter and was accompanied by an increase of longer amylopectin chains ranging from DP 45 to 100. Differences in amylopectin CLD among starches from different species and organs have already been described [44]. Such differences within the same starch extract could be explained by several parameters. First, the starch metabolism may be differently regulated in different cell types thus leading to amylopectin structural differences. The drastic disparity between stomatal guard cell and mesophyll starch metabolisms in Arabidopsis leaves illustrates this phenomenon [45]. Second, differential expression and regulation of the starch biosynthetic enzymes, especially SSs, Bes, and DBEs, during storage starch accumulation may modulate the amylopectin structure at the time of sampling. If the size of starch granules is related to their maturity state and tissue aging (i.e., small and large granules originating from immature and mature cells, respectively), one could, for example, speculate that temporal expression patterns of starch synthetizing enzymes induce amylopectin structural variations during granule growth. While the dynamics of starch synthesis and degradation were investigated at the granule level in Arabidopsis leaves [46], such analysis remains to be performed for storage starch accumulation. Furthermore, a significantly marked decrease in phosphate concentration was observed with the increase of the average granule diameter in the different starch fractions. A previous study mentioned that small potato starch granules are richer in phosphate than larger ones [47]. Here, we show that the amount of starch phosphoesters decreases with increasing granule size at both C3- and C6-position. Moreover, the decrease of total phosphate content mostly comes from the variation of C6-phosphoester concentration.

In order to relate the structural and morphological heterogeneity of starch to the corresponding metabolic pathways, we investigated the starch-bound proteome in the fractionated granule populations. Bottom-up proteomics allowed to identify all the proteins that we reported previously [16]. Label-free quantitation was performed to evaluate the concentration changes of each protein in the different fractions. Even though it is not clear whether all starch-binding enzymes are active when entrapped in the granule, such analysis provides a snapshot of the amount of each protein at the time of harvesting. Our study shows that the protein stoichiometry within starch granules significantly changes among the four fractions. Taken individually, the protein concentrations changed from less than 0.2-fold to more than 4-fold according to the average granule diameter. The most stable concentrations were those of GBSS and THRx. Conversely, PTST1 and SS3 harbored the most drastic variations in concentration. Proteins could be classified into four groups depending on their fold-change profiles which was confirmed by paired-correlation analysis. While a group of proteins comprising SS4, SS6, BE1.1, BE1.2, ISA3, GPx, and CYP20-2 showed similar patterns with a decrease followed by a continuous increase with the increase of granule diameter, SS1, 2 and 3, BE2, PTST1, LSF2, ESV1, LESV, PWD, PHS1a and b grouped together and harbored an inverse profile. These observations may illustrate the existence of protein complexes inside the starch granules such as those described in wheat, maize, and rice [48,49,50]. As expected, a strong positive correlation was observed between phosphate contents and the concentration of GWD. Although the decrease in GWD concentration was low relatively to changes of other proteins, it continuously decreased with the increase of granule size similar to the concentration of glucose-6-P. It is intuitive that the decrease in phosphate concentration results from the decreased concentration of GWD. However, correlation analysis do not allow to conclude on the causal links between the correlated parameters. This is especially important concerning the multiple potential relationships between the starch-bound proteins themselves or between the latter and starch structure or morphology. For example, the strong correlation between phosphate content and granule morphology could either result from an impact of starch phosphoesters on granule size or from a temporal variation of GWD expression during granule growth. Further investigation including single cell transcriptomics and isotopic protein labelling in pulse-chase proteomic experiments will allow to address these questions. Overall, this study corroborates that the starch-bound proteome may influence the starch granule growth and architecture thus opening new perspectives in understanding the dynamics of starch biosynthesis.

## 4. Materials and Methods

All solutions were prepared using ultrapure water purified with a MilliQ™ Academic system (Merck Millipore, Burlington, MA, USA). All chemicals, biochemicals and solvents were purchased from Sigma-Aldrich (Saint-Louis, MO, USA) and used without purification.

### 4.1. Starch Extraction

Starch granules were isolated from tubers of *Solanum tuberosum* (Monalisa) cultivated in the field in Villeneuve d’Ascq (50.607735, 3.143431), France, between March and July 2014. Potato tubers (approximately 70 g) were washed with tap water and peeled prior extraction. The tubers were ground with a blender in 200 mL of ultrapure water. The tuber extracts were then filtered through a nylon net (100 µm mesh) and left for starch granule sedimentation during 3 h. The supernatant was then removed and sedimented starch was resuspended in 500 mL of ultrapure water. The starch suspensions were subsequently washed three times with 1 L of ultrapure water and stored in 20% ethanol at 4 °C.

### 4.2. Granulomorphometry

The starch granule size and shape were measured with an Ipac 1 granulomorphometer (Occhio, Angleur, Belgium). The retained parameters for granule size were the ISO area diameter and granule shape with ellipsoid roundness. Before measurement, a numerical filter was applied. It excluded particles with a diameter smaller than 4 µm or higher than 100 µm. In addition, particles with an ellipsoid roundness lower than 46% and a diameter between 4 and 18 µm were also excluded. An average number of 3000 granules was analyzed.

### 4.3. Filter-Fractionation of Starch Granules

Granules were separated using nylon filters with a mesh width of 100, 50, 31, and 11 µm. The first filtration was realized with the 11 µm filter in order to isolate fraction A. The retained fraction was refiltered with a 31 µm filter to obtain fraction B. Then, the new retained fraction was filtered with a 50 µm filter. The eluted fraction was fraction C, and the retained fraction was fraction D. The fractionation quality was estimated by size measurement with an Ipac 1 granulomorphometer (Occhio, Angleur, Belgium). The parameters used were the same as described above.

### 4.4. Scanning Electron Microscopy (SEM)

Drops of dilute aqueous suspensions of starch granules were deposited on metallic stubs covered with C/Ni tape and allowed to dry. The specimens were coated with Au/Pd in a Baltec MED 020 sputter coater (Leica, Wetzlar, Allemagne) and observed in secondary electron mode in a FEI Quanta FEG 250 microscope (Hillsboro, OR, USA) operating at an accelerating voltage of 2.5 kV.

### 4.5. X-Ray Diffraction (XRD)

The starch granule fractions were placed in a chamber maintaining a relative humidity (R.H.) of 93% for 5 days. The powders were then poured into 0.7-mm (outer diameter) glass capillaries that were flame-sealed. All specimens were X-rayed by transmission in a Warhus vacuum chamber with a Ni-filtered CuKα radiation (*λ* = 0.1542 nm) using a Philips PW3830 generator (Eindhoven, The Netherlands) operating at 30 kV and 20 mA. Two-dimensional XRD patterns were recorded on Fujifilm imaging plates exposed for 2 h and read off-line with a Fujifilm BAS 1800-II bioimaging analyzer (Minato, Tokyo, Japan). Diffraction profiles were calculated as rotational averages of the 2D patterns.

### 4.6. Amylose Content

The protocol for amylose assay was adapted for Megazyme (Wicklow, Ireland) K-AMYL kit [51,52]. Briefly, 1.5 mg of starch from each fraction was gelatinized in 100% dimethylsulfoxide (DMSO), and precipitated with 95% ethanol in order to remove lipids. Starch was then resolubilized in 100% DMSO. Amylopectin was precipitated with concavalin A (ConA) following the manual instructions. After centrifugation, the supernatant containing amylose was recovered, and the pellet containing amylopectin was resolubilized in 100% DMSO. ConA was then inactivated by heating at 100 °C. Amylose, amylopectin, and total starch were then digested using the glucose determination reagent (GOPOD) available in the Megazyme K-AMYL kit. The absorbance was read at 510 nm using a SPECTROstar Nano spectrometer (BMG LABTECH’s, Allmendgrün, Germany). At the same time, an iodine staining was performed on amylose, and amylopectin fractions, and total starch. The absorbance was read between 500 and 700 nm in order to determine the wavelength at the maximum of absorbance (*λ_max_*) for each glucan-I_2_ complex.

### 4.7. Amylopectin Chain Length Distribution

The amylopectin chain length distribution was measured by fluorescence-assisted capillary electrophoresis (FACE). Two milligram of starch from each fraction, in suspension in 250 µL of ultrapure water, was gelatinized by heating at 100 °C, with intermittent high-speed stirring in a vortex mixer, until the solution became clear. Then, 20 U of isoamylase and 1 U of pullulanase were added and the solution was incubated at 42 °C overnight in order to debranch amylopectin. Proteins were then precipitated by heating at 95 °C for 15 min, and the supernatant was recovered after centrifugation. The samples were desalted on Alltech™ Carbograph (Grace, Columbia, MD, USA) solid-phase extraction (SPE) column, then freeze-dried. The dry samples were resolubilized in 2 µL of 1 M sodium cyanoborohydride in tetrahydrofuran (THF), and 2 µL 0.2 M 8-aminopyrene-1,3,6-trisulfonic acid trisodium salt (APTS) in 15% acetic acid (*v*/*v*). The samples were incubated overnight at 42 °C, and diluted with 46 µL ultrapure water. After the first dilution, the sample was then rediluted 50 times with ultrapure water prior to injection in a Beckman Coulter PA800-plus Pharmaceutical Analysis System (Beckman Coulter, Brea, CA, USA) equipped with a silicon capillary column (inner diameter: 50 µm; outer diameter: 360 µm; length: 60 cm). The capillary was rinsed and coated with Beckman Coulter *N*-linked carbohydrate separation gel buffer one-third diluted with ultrapure water prior to injection (15 s at 10 kV). The migration was performed at 10 kV during 1 h at 20 °C. The detection was made with a 488 nm solid-state laser module, and a laser-induced fluorescence (LIF) detector. The detection window was located 10 cm above the outlet of the capillary column.

### 4.8. Phosphate Content

The phosphate content for each fraction was determined by FACE. The protocol was the same as that described in Verbeke et al. 2016. Briefly, 1 mg of starch from each fraction was submitted to acid hydrolysis with 100 µL of 2 M trifluoroacetic acid (TFA) at 95 °C during 1 h and 20 min. A total of 20 µL of the hydrolysate was then diluted with 200 µL of ultrapure water to stop the reaction. The sample was then dried with a SpeedVac™ Concentrator (Eppendorf™ Concentrator Plus, Eppendorf, Hamburg, Germany). The dry samples were then derivatized with APTS as described before. The derivatized samples were diluted with 96 µL of ultrapure water. A total of 15 µL of this first dilution was rediluted with 185 µL of ultrapure water prior to injection in a Beckman Coulter PA800-plus Pharmaceutical Analysis System with the same silicon capillary column as that described above. The capillary was rinsed and coated with the Beckman Coulter *N*-linked carbohydrate separation gel buffer prior to injection (10 s at 10 kV). The migration was performed at 10 kV during 10 min. The detection was made at 488 nm as described above.

### 4.9. Starch Granule Associated Protein Extraction

The starch proteins were extracted from 300 mg of dry fractionated starch. Ten aliquots of 30 mg starch were then suspended in 500 µL of extraction buffer (0.2 M Tris at pH 6.8; 2% sodium dodecyl sulfate (SDS); 50 mM dithiotreithol (DTT). Aliquots are then heated to 100 °C until gelatinization of the granules. After gelatinization, the samples were kept for an additional 10 min at 100 °C and centrifuged for 10 min at 13,000 rounds per minute (rpm). Supernatants containing proteins were rescued, pooled, and dropped on an Amicon^®^ (Millipore, Billerica, MA, USA) membrane filter with a 50 kDa cutoff inside a 15 mL vial. The vial was centrifuged at 7500 rpm for 30 min in order to concentrate the samples. A total of 5 µL of each sample, diluted 10 times, was used to carry out a Bicinchoninic assay (BCA; Thermo Fisher Scientific, Waltham, MA, USA) following the manual instructions.

### 4.10. Label-Free Quantitative Proteomics, Sample Preparation

A total of 50 µg of proteins are reduced, alkylated, and digested with eFASP Digestion protocol. Ultra-filtration filters from Amicon^®^ units (10 kDa cutoff limit; Millipore, Billerica, MA, USA) were incubated overnight with 5% *v*/*v* TWEEN-20 (T20, Sigma-Aldrich, Saint-Louis, MO, USA). Following incubation, the filter units were rinsed thoroughly by three immersions in ultrapure water. eFASP digestion is realized with 50 µg of samples estimated from the BCA assay. These samples were mixed in 50 µL of solubilization and reducing buffer A (4% SDS, 0.2% deoxycholic acid (DCA), 50 mM DTT, 200 mM ammonium bicarbonate (ABC)) overnight at 4 °C. Samples were centrifuged at 13,000 *g* at 20 °C for 15 min. Supernatants were mixed with 200 µL of SDS buffer exchange (8 M urea, 0.2% DCA, 100 mM ABC, pH 8) and were transferred into Amicon^®^ units. After 30 min of 13,000 *g* centrifugation, the filtrate was discarded, an additional 200 µL of exchange buffer A was deposited in each unit, and centrifugation resumed for 30 more minutes. This buffer addition/centrifugation step was repeated twice more. The reduced proteins were alkylated within the filter unit by adding alkylation buffer A (8 M urea, 50 mM iodoacetamide, and 100 mM ABC, pH 8) and incubating at 37 °C for 1 h with shaking in the dark. After 30 min of 13,000 *g* centrifugation, the filtrate was discarded, an additional 200 µL of exchange buffer A was deposited in each unit, and centrifugation resumed for 30 more minutes. This buffer addition/centrifugation step was repeated once more. After buffer exchange was performed with three exchanges with eFASP digestion buffer (50 mM ABC, 0.2% DCA pH 8), 100 µL of eFASP digestion buffer was added to each Amicon^®^ units, followed by 1 µg of trypsin (1:50 w/w). Digestion proceeded for 16 h at 37 °C. Peptides were recovered by transferring the UF filter to a new collection tube and spinning at 13 000 *g* for 20 min. To complete peptide recovery, filters were rinsed twice with 50 µL of 50 mM ABC that was collected by centrifugation. A total of 200 µL of ethyl acetate was added to the peptide containing filtrate and was transferred to a 2 mL tube. After that 2.5 µL TFA was added and was quickly vortex. White thread like precipitate may be visible for large quantities of peptides. Peptides precipitate were mixed with 800 µL of ethyl acetate and were centrifuged at 13,000 *g* for 10 min. The organic supernatant was discarded and this step was repeated twice more. The upper organic layer was carefully removed and discarded as much as possible without disturbing the boundary layer. Uncovered sample tubes were placed in a thermomixer at 60 °C, in a fume hood, for 5 min to remove residual ethyl acetate. Residual organic and volatile salts were removed by vacuum drying in a SpeedVac™ Concentrator (Eppendorf™ Concentrator Plus, Eppendorf, Hamburg, Germany). This step was repeated two times with 50% methanol. Ten micrograms of dry peptides are then resuspended in 0.1% (*v*/*v*) formic acid.

### 4.11. Mass Spectrometry Analysis

A nanoflow HPLC instrument (U3000 RSLC, Thermo Fisher Scientific, Waltham, MA, USA) was coupled on-line to a Q Exactive plus Orbitrap mass spectrometer (Thermo Fisher Scientific) fitted with a nanoelectrospray ion source. One microliter of peptide mixture (corresponding to 500 ng of proteins) was loaded onto the preconcentration trap (Thermo Fisher Scientific, Acclaim PepMap100 C18, 5 µm, 300 µm i.d × 5 mm) using a partial loop injection, for 5 min at a flow rate of 10 µL.min^−1^ with buffer A (5% acetonitrile and 0.1% formic acid) then separated on column (Acclaim PepMap100 C18, 3 µm, 75 mm i.d. × 500 mm) and separated with a linear gradient of 5–50% buffer B (75% acetonitrile and 0.1% formic acid) at a flow rate of 250 nL.min^−1^ and temperature of 45 °C. The total time for an LC MS/MS run was 270 min. MS data was acquired on Q Exactive plus Orbitrap mass spectrometer (Thermo Fisher Scientific) using a data-dependent top 20 method, dynamically choosing the most abundant precursor ions from the survey scan (350–1600 m/z) for HCD fragmentation. Dynamic exclusion duration was 60 s. Isolation of precursors was performed with a 1.6 m/z window and MS/MS scans were acquired with a starting mass of 80 m/z. Survey scans were acquired at a resolution of 70,000 at m/z 400 (AGC set to 10^6^ ions with a maximum fill time of 100 ms). Resolution for HCD spectra was set to 70,000 at m/z 200 (AGC set to 10^5^ ions with a maximum fill time of 200 ms). Normalized collision energy was 28 eV. The underfill ratio, which specifies the minimum percentage of the target value likely to be reached at maximum fill time, was defined as 0.4%. The instruments were run with peptide recognition mode (i.e., from 2 to 8 charge), exclusion of singly charged and of unassigned precursor ions enabled.

### 4.12. Proteins Label Free Quantification with Maxquant and Perseus

Proteins were quantified with Maxquant software 1.5.3.30 with custom starch bound protein databank (Appendix A) regrouping proteins identified as bound to starch in our previous study [16], in order to obtain iBAQ and LFQ (label-free quantitation) score for each identified proteins. iBAQ score is used on Perseus 1.6.0.2 software to obtain the abundance of each protein. Retained parameter for the quantification is “Concentration [nM] iBAQ” (Appendix A). The concentration score is then divided by the total concentration of protein in µg.mg of starch^−1^ in order to obtain a result in fmol.mg of starch^−1^. Fold change values are obtained by normalizing the concentration of one protein on the concentration of this protein in the fraction A. LFQ values are transformed in log2 then filtered and used on Perseus 1.6.0.2 to realize statistical testing. For multiple comparisons, ANOVA analysis with a permutated based FDR correction (FDR = 0.05) was performed. For the comparison between two groups, *t*-tests were performed with P-values less than 0.05 were considered to be statistically significant (Appendix A). The correlations between variables were defined by the Pearson coefficient and performed using R version 3.6.0.

## Figures and Tables

**Figure 1 plants-08-00324-f001:**
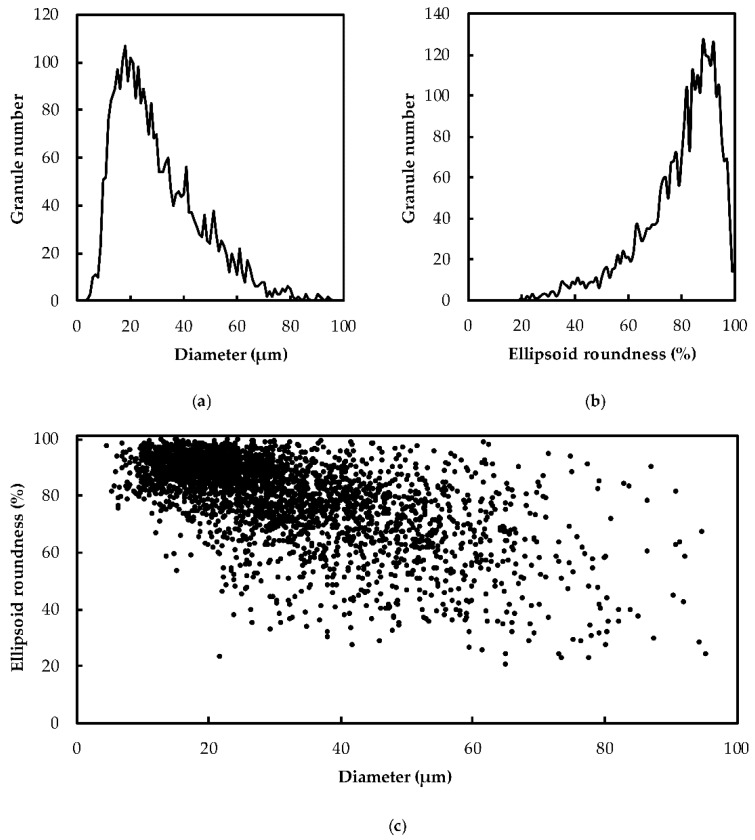
Size and shape analysis of unfractionated starch granules from potato cv. Monalisa. About 3000 starch granules were used for granulomorphometry. (**a**) Size distribution of potato starch. The number of analyzed granules was plotted versus their corresponding area diameter. (**b**) Shape distribution of Monalisa potato starch. Same as in (a) with the use of the ellipsoid roundness parameter. (**c**) Scatter plot of the ellipsoid roundness versus granule diameter.

**Figure 2 plants-08-00324-f002:**
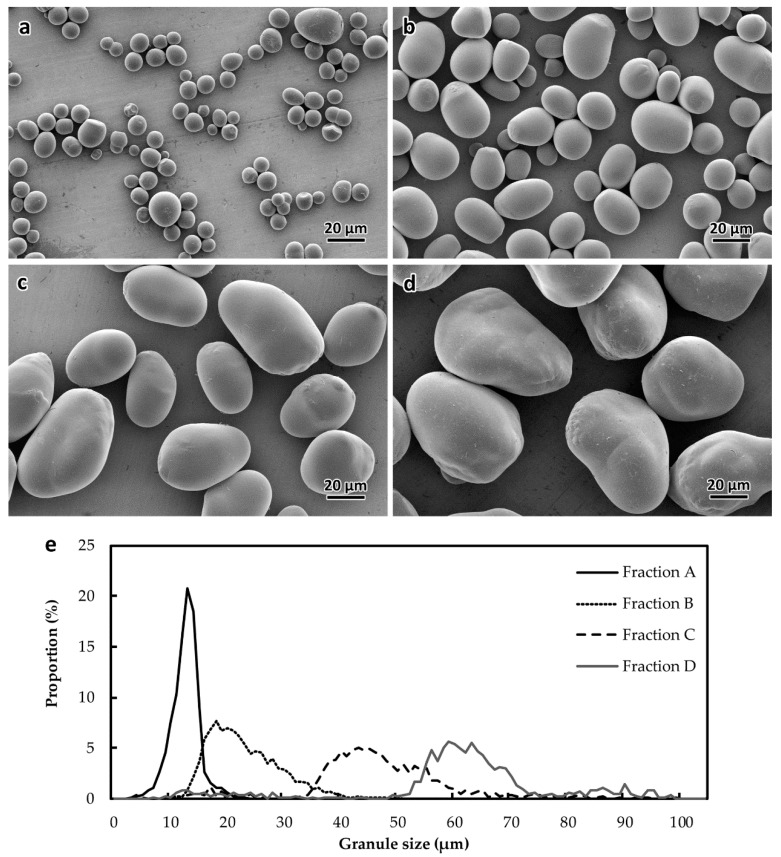
Morphology of size-fractionated starch granules. Scanning electron microscopy images of fractions A (**a**), B (**b**), C (**c**), and D (**d**). (**e**) Size distribution of each fraction. 3035 granules from fraction A, 3436 granules from fraction B, 1198 granules from fraction C, and 784 granules from fraction D were used for granulomorphometry.

**Figure 3 plants-08-00324-f003:**
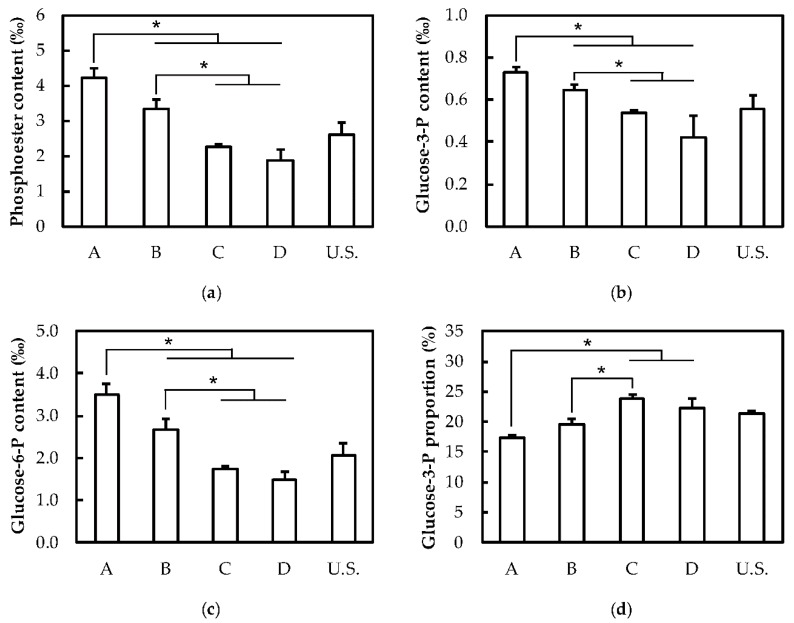
Phosphate contents. One milligram of starch from fractions A, B, C, and D and unfractionated starch (U.S.) was hydrolyzed prior to APTS-labelling and capillary electrophoresis analysis. (**a**) Total phosphate contents. (**b**) Glucose-3-P contents. (**c**) Glucose-6-P contents. (**d**) Proportion of glucose-3-P expressed in % of total phosphate. Error bars represent SDs calculated from three experimental replicates. The asterisks indicate significant differences according to one-way ANOVA with a P-value ≤ 0.01.

**Figure 4 plants-08-00324-f004:**
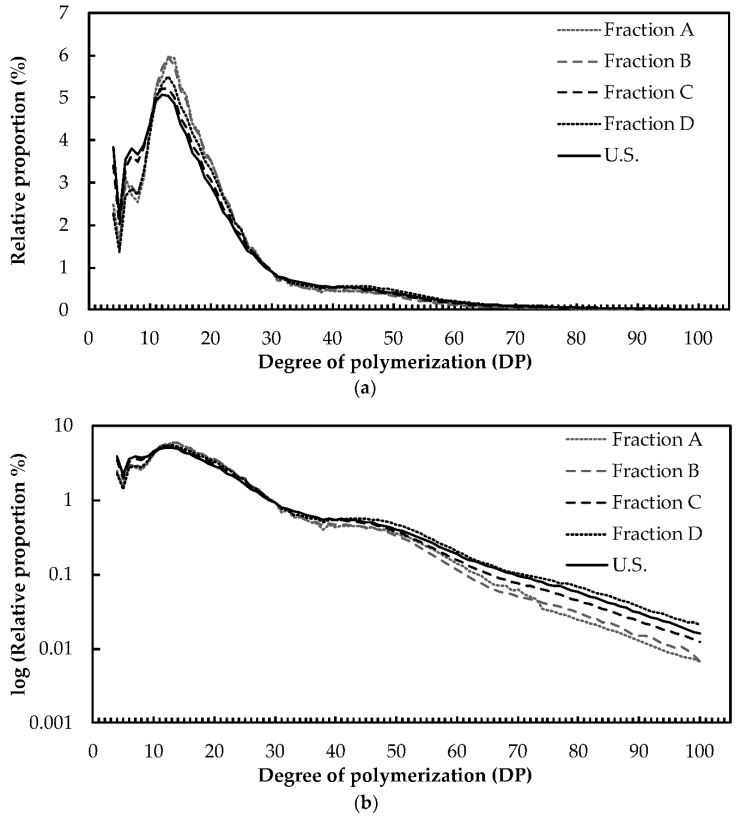
**Amylopectin chain length distribution.** One milligram of starch from each fraction and unfractionated starch (U.S.) was derivatized with APTS after enzymatic digestion by pullulanase and isoamylase. Peaks corresponding to DP 4 to DP 100 were integrated and their relative proportion in % is displayed. (**a**) Chain length distribution in linear scale. (**b**) Chain length distribution in log_10_ scale. Glucan chain proportion was averaged from analyses of two independent fractionations.

**Figure 5 plants-08-00324-f005:**
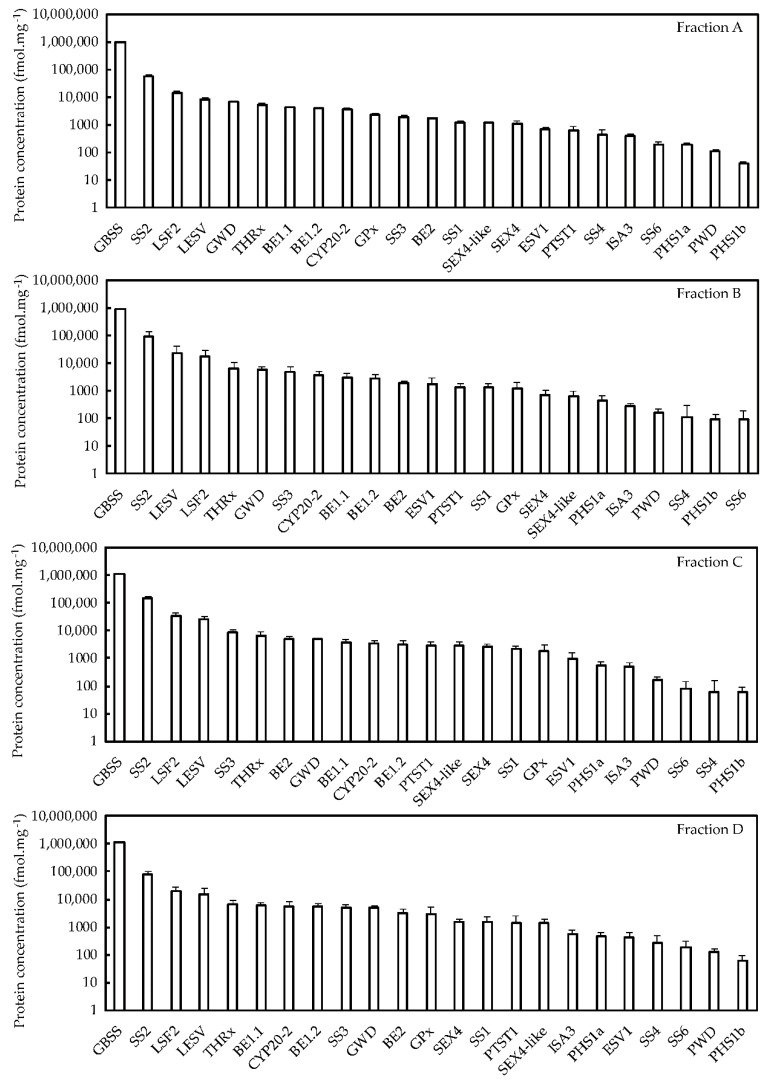
Label-free quantitative proteomics of each fraction. Proteins were isolated from 300 mg of starch. They were then digested into peptides and analyzed by MS/MS. Interrogation was realized with Maxquant and Perseus software. Protein concentration in nM was calculated with the Perseus algorithm using the iBAQ score from Maxquant. These concentrations were then expressed in fmol and normalized to starch quantity (mg) using the protein concentrations displayed in Table 1. Error bars represent SDs calculated from three experimental replicates, with three technical replicates for each. Statistical analysis was performed by ANOVA using the LFQ scores from the Maxquant analysis (Appendix A).

**Figure 6 plants-08-00324-f006:**
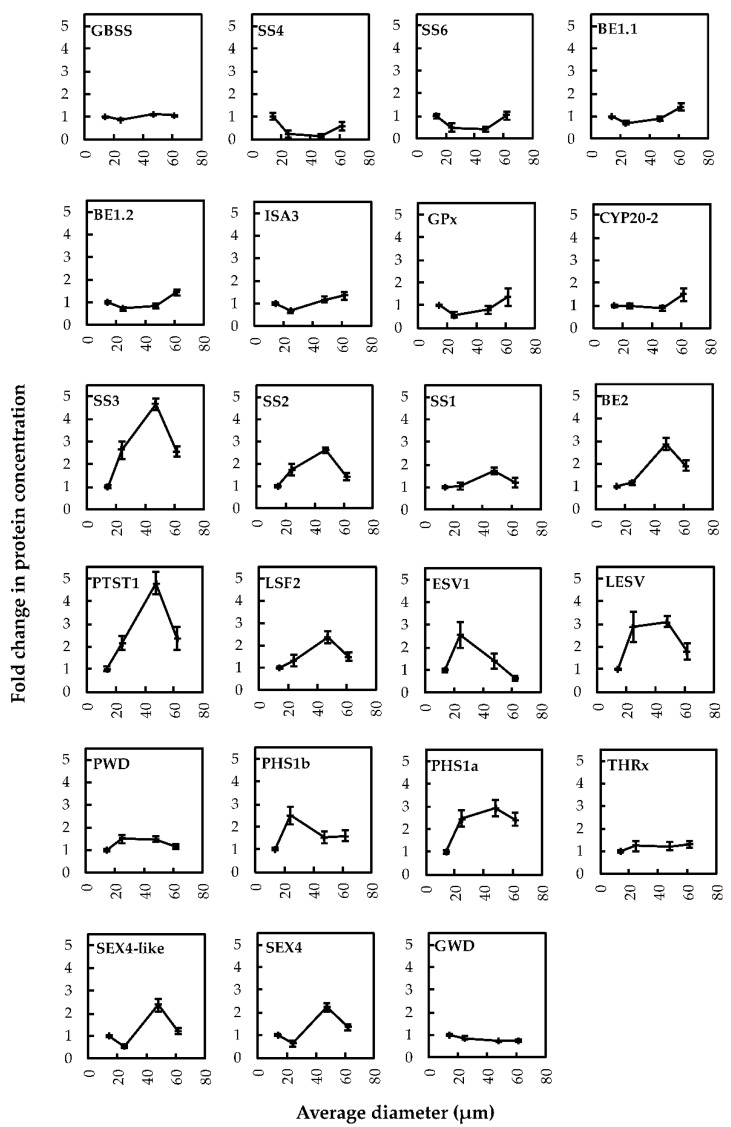
Fold changes in protein concentrations according to average granule diameter. Fold changes were calculated by normalizing the protein content in fraction A to 1. Error bars represent standard errors (SEs) calculated from three experimental replicates, with three technical replicates for each.

**Table 1 plants-08-00324-t001:** Size, shape, protein and amylose contents and *λ_max_* of the amylopectin-I_2_ complex. Statistics were performed by Student t-test compared to unfractionated starch (U.S.). * corresponds to P < 0.05 and ** to P < 0.001.

Fraction	Average Diameter (µm)	Ellipsoid Roundness (%)	Protein Content (µg·mg^−1^)	Amylose Content (%)	*λ_max_* of the Amylopectin-I_2_ Complex (nm)
A	14.2 ± 3.7 **	85.9 ± 12.5 **	2.6 ± 0.2 *	22.6 ± 5.1	548 ± 2
B	24.5 ± 6.5 **	82.5 ± 13.6 **	2.8 ± 0.1 **	20.7 ± 2.1	546 ± 4
C	47.7 ± 12.8 **	72.0 ± 14.8 **	2.1 ± 0.4 *	20.8 ± 2.9	552 ± 2
D	61.8 ± 17.4 **	73.2 ± 15.0 **	2.4 ± 0.1 *	20.6 ± 1.9	549 ± 2
U.S.	30.7 ± 15.9	79.9 ± 14.6	2.5 ± 0.2	21.5 ± 1.0	554 ± 4

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
