# Peer review of "Intra-Sample Heterogeneity of Potato Starch Reveals Fluctuation of Starch-Binding Proteins According to Granule Morphology"

_plants, 2019, doi:10.3390/plants8090324_

Round 1

Reviewer 1 Report

The manuscript by Helle et al describes the relationship between starch granule size and starch binding proteins in potato tuber. I find the experiments neatly designed and described, and the results presented clearly. The manuscript is well written with an easy to read story flow. The results expand our current understanding of the starch binding proteins in starch synthesis and granule structure/formation. Overall, I recommend publication of this manuscript. 

Author Response

We wish to thank you for the time and effort that you have put into assessing the previous version of the manuscript as well as for your positive feedback.

Reviewer 2 Report

In this manuscript, researchers investigated the fluctuation of starch-binding proteins based on the granule morphology by studying the intra-sample heterogeneity. Though there are many researches to reveal the starch synthesis process in plans, still many mechanisms are unknown. The current study clearly suggested that the various starch-binding proteins effects the form of starch granule during the biosynthesis by studying the starch sizes, amylose contents, and other structural properties of different granule-sizes of potato starch. The manuscript is well written and discussed about topic with supporting citations and explanations. Thus, it can be publishable in Plants.

Lines 89-92: need a brief research objective

Line 188, 288, and Table 3: Use full words (Glc -> glucose) though those are widely known abbreviations.

Author Response

We wish to thank you for the time and effort that you have put into assessing the previous version of the manuscript, as well as for the careful and constructive review. We now briefly state the research objective (Lines 89–91): “In order to evaluate the potential heterogeneity of starch-binding proteins in link with starch structure and morphology, we investigated the size and shape repartition as well as the composition and structure of size-fractionated potato starch granule populations…”. We also replaced “Glc” by “glucose” (Fig.3, Lines 162, 189, 289).

Reviewer 3 Report

The manuscript requires the following corrections before publication:

-Figure 1C: improve the resolution.

-Figure 2C: improve the resolution.

-Figure 3 (a-d): improve the resolution.

-Figure 4(a-b): improve the resolution.

-Figure 3 (a-d): In the figure is not reported any indication about the statistical significance of data that is instead reported in the related caption.

Author Response

We wish to thank you for the time and effort that you have put into assessing the previous version of the manuscript, as well as for the careful and constructive review. We now provide with Fig. 1, 2e, 3 and 4 at a resolution of 1200 dpi. Asterisks were added in Fig. 3 to indicate statistical differences. The caption was modified accordingly (Lines 164-165).